# Chemical composition and light absorption of carbonaceous aerosols emitted from crop residue burning: Influence of combustion efficiency

Yujue Wang,[1] Min Hu,[*,1,2,4] Nan Xu,[1] Yanhong Qin,[1] Zhijun Wu,[1,2] Liwu Zeng,[3] Xiaofeng Huang,[3] Lingyan He[3]

[1]State Key Joint Laboratory of Environmental Simulation and Pollution Control, College of Environmental Sciences and Engineering, Peking University, Beijing 100871, China

[2]Collaborative Innovation Center of Atmospheric Environment and Equipment Technology, Nanjing University of Information Science & Technology, Nanjing, China

[3]Key Laboratory for Urban Habitat Environmental Science and Technology, School of Environment and Energy, Peking University Shenzhen Graduate School, Shenzhen, China

[4]Beijing Innovation Center for Engineering Sciences and Advanced Technology, Peking University, Beijing 100871, China

*Correspondence to*: Min Hu (minhu@pku.edu.cn)

**Abstract.** Biomass burning is one of the major sources of carbonaceous aerosols, which affects air quality, radiation budget and human health. Field straw residue burning is a widespread type of biomass burning in Asia, while its emissions are poorly understood compared with the wood burning emissions. In this study, lab-controlled straw (wheat and corn) burning experiments were designed to investigate the emission factors and light absorption properties of different biomass burning organic aerosol (BBOA) fractions, including water soluble organic carbon (WSOC), humic-like substances (HULIS) and water insoluble organic carbon (WISOC). The influences of biofuel moisture content and combustion efficiency on emissions are comprehensively discussed. The emission factors of $PM_{2.5}$, OC and EC were 9.3±3.4, 4.6±1.9 and 0.21±0.07 g/kg for corn burning and 8.7±5.0, 3.9±2.8 and 0.22±0.05 g/kg for wheat burning, generally lower than wood or forest burning emissions. Though the mass contribution of WISOC among OC (32%-43%) was lower than WSOC, the light absorption contribution of WISOC (57%−84% @300-400 nm) surpassed WSOC due to the higher mass absorption efficiency (MAE) of WISOC. The results suggested that BBOA light absorption would be largely underestimated if only considering the water soluble fractions. However, the light absorption of WSOC among near-UV ranges, occupying 39%-43% of the total extracted OC absorption at 300 nm, cannot be negligible due to the sharper increase of absorption towards shorter wavelength compared with WISOC. HULIS were the major light absorption contributors among WSOC, due to the higher MAE of HULIS than other high-polarity WSOC components. The emission levels and light absorption of BBOA were largely influenced by the burning conditions, indicated by modified combustion efficiency (MCE) calculated by measured CO and $CO_2$ in this study. The emission factors of $PM_{2.5}$, OC, WSOC, HULIS and organic acids were enhanced under lower-MCE conditions or during higher-moisture straw burning experiments. Light absorption coefficients of BBOA at 365

nm were also higher under lower-MCE conditions, which was mainly due to the elevated mass emission factors. Our results suggested that the influence of varied combustion efficiency on particle emissions could surpass the differences caused by different types of biofuels. Thus, the burning efficiency or conditions should be taken into consideration when estimating the influence of biomass burning. In addition, we observed that the ratios of $K^+$/OC and $Cl^-$/OC increased under higher-MCE conditions due to the enhancement of released potassium and chlorine under higher fire temperatures during flaming combustion. This indicates that potassium ion, as a commonly used biomass burning tracer, may lead to estimation uncertainty without considering the burning conditions.

## 1 Introduction

Biomass burning emissions, as a major primary source of carbonaceous aerosols, have significant effects on the air quality, human health as well as regional or global radiation budget (Bond, 2004; Chen et al., 2017a; Reid et al., 2005; Saleh et al., 2015). Biomass burning could contribute one-third of the black carbon (BC) budget and two-thirds of the primary organic aerosol budget on the global scale (Bond, 2004; Bond et al., 2013). In recent years, biomass burning organic aerosols (BBOA) also attracted much attention due to their substantial contribution to light-absorbing organic aerosols, known as brown carbon (BrC) (Andreae and Gelencsér, 2006; Laskin et al., 2015; Lin et al., 2016; Saleh et al., 2014; Washenfelder et al., 2015; Yan et al., 2018). Emission factors (EF) of BrC ranged from 1.0 to 1.4 g/kg biomass, comparable to those of BC (Aurell and Gullett, 2013). Majority of BrC aerosol mass was associated with biomass burning emissions in rural southeast US (Washenfelder et al., 2015). Regional radiative forcing effects of BrC could be comparable to those of BC over major areas dominated by biomass burning and biofuel combustion, such as South and East Asia (Feng et al., 2013).

Emission factors, chemical compositions and light absorption properties of biomass burning aerosols could be obviously influenced by different types of biomass, biofuel structures, moisture contents, and especially varied burning conditions (Chen and Bond, 2010; Holder et al., 2016; Reisen et al., 2018). The emissions of particulate organics could span several orders of magnitude depending on different burning conditions (Chen et al., 2017a; Jen et al., 2019). In general, higher levels of particulate matters (PM) and organic aerosols were emitted during less efficient biomass burning, due to prolonged incomplete or smoldering combustion (Holder et al., 2016; Jen et al., 2019; Reisen et al., 2018). Open biomass burning, especially smoldering combustion, dominates the organic carbon (OC) emissions in many regions of the world on an annual-average basis (Bond, 2004). The light absorption of biomass burning aerosols are also largely dependent on the combustion conditions (Cheng et al., 2016a; Liu et al., 2014; Pokhrel et al., 2016; Saleh et al., 2014). The contribution of BrC to aerosol light absorption at near-UV wavelength was reported to be higher for more smoldering combustion compared with more flaming combustion (Holder et al., 2016). The reported variation trends of BBOA absorption properties as a

function of combustion conditions, however, are not consistent from different studies. High variability in reported emission factors and optical properties of BBOA from different burning conditions complicates their treatment in climate models (Liu et al., 2014; Saleh et al., 2014), and indicates the importance of further investigations on biomass burning emissions, especially the influence of burning conditions.

Unlike the well-understood BC, the light-absorbing OC or BrC comprise a wide range of poorly characterized organic compounds, which exhibit highly variable chemical and light absorption properties (Andreae and Gelencsér, 2006; Laskin et al., 2015; Lin et al., 2016; Saleh et al., 2014; Washenfelder et al., 2015; Yan et al., 2018). Previous studies have suggested that methanol extracted BrC were usually more light-absorbing than water extracts for BBOA or ambient aerosols (Chen and Bond, 2010; Liu et al., 2013). More than 92% of the light absorbing OC emitted from solid fuel pyrolysis could be extracted by methanol, compared with 73% for water-extracted compounds (Chen and Bond, 2010). Alkaline or methanol extracted OC fractions were also observed with higher mass absorption efficiency (MAE) at 365 nm than water soluble organic carbon (WSOC) for residential coal combustion (Li et al., 2018). Only considering the water soluble BrC would result in underestimation of BrC absorption and radiative forcing (Cheng et al., 2016b; Cheng et al., 2017). Different light absorption properties of organic fractions could be attributed to the varied chemical compositions and structures (Chen et al., 2016a; Chen et al., 2016b; Chen et al., 2017b). However, few studies have been conducted to gain a comprehensive understanding on the influence of combustion conditions on the chemical composition and light absorption of different BBOA fractions from agricultural residue burning.

Field open burning of agriculture wastes or crop residues is a widespread type of biomass burning in Asia (IARI, 2012; Bond, 2004; Streets et al., 2003a). Open crop residue burning during harvest season would result in severely adverse impacts on regional air quality and human health (Chen et al., 2017a; Li et al., 2014; Lin and Yu, 2011; Streets et al., 2003b; Zhang et al., 2010). The PM emission factors from agricultural waste burning range from 1.7 to 17.8 g/kg (Bond, 2004). Source apportionment results showed that ~50% of carbonaceous aerosols in Beijing were associated with biomass burning, with crop residue combustion as a major source (Cheng et al., 2013). Straw residue burning could contribute as high as 51% of PM and 76% of OC during harvest seasons in the agriculture regions in China (Li et al., 2014). Considering the large contribution of straw residue burning, the chemical compositions and light absorption properties of BBOA in Asia may differ from other regions with wood burning as the major type of biomass burning. However, the understanding on field straw residue burning emissions is still limited. A better characterization of the emission levels and optical properties of straw burning aerosols is required to quantify their effects on air quality and regional radiation forcing in agriculture areas (Hungershoefer et al., 2008). Laboratory simulation experiment has been suggested as a good way to study biomass burning emissions due to its advantage in quantifying emission factors and controlling combustion conditions within well-defined limits. In this study, a series of lab burning experiments were designed to systematically investigate the emission factors,

chemical compositions and light absorption properties of both water-soluble and water-insoluble carbonaceous aerosols
emitted from straw residue burning. The influence of biofuel moisture contents, burning conditions and combustion
efficiency on the BBOA emission levels and light absorption properties are comprehensively discussed.

## 2 Methods

### 2.1 Simulation and sampling of biomass burning aerosols

Lab-controlled burning experiments were conducted in the Laboratory of Biomass Burning Simulation at Peking
University Shenzhen Graduate School. The simulation system was designed and optimized on the basis of the one used in He
et al. (2010) (He et al., 2010), which included combustion system, dilution system, sampling system and data acquisition
system (Figure S1). During each experiment, about 1-2 kg biomass fuels were ignited on the combustion pan. The emitted
smoke was collected by the hood above the fire, and diluted by zero air (21 mol% $O_2$ and 79 mol % $N_2$) before collected on
filters or monitored by online instruments. Smoke aerosols were collected on both Teflon (Whatman Inc.) and quartz fiber
(Whatman Inc.) filters, using a $PM_{2.5}$ cutoff with a sampling flow rate of 16.7 L/min. During each burning experiment, CO
and $CO_2$ were measured continuously by CO and $CO_2$ analyzers (Thermo Scientific Inc., Bremen, Germany). The burning
efficiency, calculated based on the online CO and $CO_2$ data, were monitored continuously during each experiment (Table S1).
The variation of fire temperatures during each experiment was also measured by a sensor above the fire (Figure S1).
In this study, corn and wheat, two kinds of primary grain crops in China, burning was simulated to represent the straw
residue burning in China. To investigate the influence of biofuel moisture contents on burning emissions, straws with
different levels of moisture contents were burned, including low (13%) and high (18%) levels for corn burning experiments,
low (7%-9%), medium (18%-22%), and high (27%-33%) levels for wheat burning experiments (Table S1). The moisture
content was measured by weighing the fuels before and after drying the biofuels in the oven at 105℃ for 24 h. Straw
residues with different moisture contents were prepared by mixing weighed biofuels with weighed pure water in a plastic
box, and shaking until the water was absorbed. Each experiment condition was repeated three times. All the conducted
experiment conditions as well as burning conditions are summarized in Table S1.

### 2.2 Isolation of carbonaceous aerosols

The quartz fiber filters were used to extract different carbonaceous aerosol fractions, including water-insoluble organic
carbon (WISOC), WSOC, and carbon component of HUmic-Like Substances ($HULIS_C$). The filter samples were firstly
extracted in an ultrasonic bath twice using 10 mL, and 10 mL ultrapure water, each time for 30 min. The extracts were then
combined and filtered with a 0.45 μm pore size syringe filter (Gelman Sciences) to obtain the WSOC solutions. After
removing the WSOC fraction on filters, the WISOC fractions were then extracted in an ultrasonic bath twice using 5 mL,
and 5 mL methanol, each time for 30 min. The extracts then were combined and filtered using a 0.25 μm syringe filter. The
HULIS fraction was isolated from the WSOC solutions via solid phase extraction (SPE), with majority of low molecular
weight organic acids (with relatively higher polarities) and sugars removed from the water solutions. Details about the
HULIS extraction procedures were described in our previous paper (Wang et al., 2017). The WSOC fraction excluded
HULIS was named as high-polarity WSOC (WSOC-h) in this study.
**2.3 Quantification and light absorption measurements of carbonaceous aerosols**
The total OC abundance was analyzed by a thermal/optical carbon analyzer (Sunset Laboratory). The concentrations of
water soluble carbonaceous aerosol fractions, including WSOC and $HULIS_C$, were measured using a total organic carbon
(TOC) analyzer (AnalytikJena multi N/C 3100). The WISOC concentrations were obtained by the difference between total
OC and WSOC. Light absorption of the extracted solutions (WSOC, $HULIS_C$ and WISOC) were measured by a UV-vis
spectrometer (UV-1780, Shimadzu) over the wavelength range of 300-700 nm. The absorptions of WSOC and WISOC were
added up to represent the absorption of the total extracted OC. The absorption coefficients ($Abs_\lambda$, $Mm^{-1}$) and mass absorption
efficiency ($MAE_\lambda$) of isolated solutions at a wavelength λ were calculated as follow (Cheng et al., 2011; Cheng et al.,
2016b):

$$Abs_\lambda = (A_\lambda - A_{700}) \frac{V_{sol}}{V_{air} \times L} \times \ln(10) \tag{1}$$

$$MAE_\lambda = \frac{Abs_\lambda}{C} \tag{2}$$

where $A_\lambda$ and $A_{700}$ represent the measured absorbance at wavelength λ and 700 nm. $A_\lambda$ is referenced to the $A_{700}$ to account for
systematic baseline drift (Xie et al., 2019; Zhang et al., 2013). $V_{sol}$ is the volume of extracted solutions and $V_{air}$ is the volume
of air sampled through the filter punch. The optical path length (L) is 1 cm in the present experiments. Ln (10) is used to
convert from common logarithm to natural logarithm. C corresponds to the concentrations of OC, WISOC, WSOC or
$HULIS_C$ fractions. It is noted that the total OC was used to represent the concentration of total extracted OC, which may lead
to an underestimation of MAE of WISOC. Previous studies suggested that 92%-99.7% of BBOA could be extracted by
methanol (Chen and Bond, 2010; Xie et al., 2019), thus the residue OC un-extracted by methanol was relatively small
compared with the extracted fraction. The wavelength dependence of light absorption is described using the Absorption
Angstrom Exponent (AAE), which is calculated by a linear regression fit of log($Abs_\lambda$) versus log(λ) in the wavelength range
of 300-450 nm.
The radiation effects of different BrC fractions (WSOC, HULIS and WISOC) relative to elemental carbon (EC, f) were
estimated using a simplified model (Kirillova et al., 2014; Wu et al., 2020):
$$f = \frac{\int I_0(\lambda)\left\{1 - e^{-\left(MAE_{BrC,365}\left(\frac{365}{\lambda}\right)^{AAE}\cdot C_{BrC}\cdot h_{ABL}\right)}\right\}d\lambda}{\int I_0(\lambda)\left\{1 - e^{-\left(MAE_{EC,870}\left(\frac{870}{\lambda}\right)\cdot C_{EC}\cdot h_{ABL}\right)}\right\}d\lambda} \qquad (3)$$

where $MAE_{BrC,365}$ and $MAE_{EC,870}$ represent the MAE of different BrC fractions at 365 nm and MAE of EC at 870 nm. AAE
is the AAE values of different BrC fractions obtained in this study, and the AAE of EC is set to 1. $C_{BrC}$ and $C_{EC}$ are the
concentrations of BrC and EC, and $h_{ABL}$ is the height of atmospheric boundary layer (1000 m). $I_0(\lambda)$ represents the clear sky
Air Mass 1 Global Horizontal solar irradiance (Levinson et al., 2010).
Water-soluble $K^+$, $Cl^-$ and low molecular weight organic acids (acetic acid, formic acid, succinic acid, oxalic acid,
propionic acid and methanesulfonic acid) were analyzed by ion chromatograph (DIONEX, ICS2500/ICS2000), following the
procedures described in Guo et al. (2010) (Guo et al., 2010).

## 3 Results and discussion

### 3.1 Burning conditions and combustion efficiency

The burning conditions and combustion efficiency, calculated by measured CO and $CO_2$ concentrations, of the
simulation experiments are shown in Figure 1 and Table S1. Modified combustion efficiency (MCE), defined as
$\Delta CO_2/(\Delta CO_2+\Delta CO)$, is used to indicate the burning conditions during a fire (Akagi et al., 2011; Andreae and Merlet, 2001).
The burning conditions in this study varied from different fires, with the MCE ranging from 0.68 to 0.88 and an average
value of 0.77. The amount and compositions of substances emitted from a given fire are determined to a large extent by the
burning conditions or the ratio of flaming to smoldering combustion, which is often expressed as "combustion efficiency".
Higher MCE (>0.9) indicates more flaming combustions, and lower MCE indicates more smoldering conditions. A previous
study suggested that pure flaming has an MCE near 0.99, and the MCE of most smoldering combustion is around or lower
than 0.8 (Akagi et al., 2011). The burning experiments were generally dominated by smoldering combustions in the present
study. Smoldering-dominated conditions, with expected MCE<0.9 or even lower, have been widely observed during the
combustion of agricultural residues in the field (IARI, 2012; Wang et al., 2017), thus the results in this study are applicable
to the field or related model studies.
The biomass fuels with lower moisture contents are generally burned more efficiently, with relatively higher MCE
values (Table S1, Figure S2), which suggested higher proportion of flaming combustion during the fire. The MCE of
higher-moisture biomass burning was generally lower, and prolonged smoldering combustion was observed (Figures 1, S2).
Previous lab-controlled burning experiments also reported similar phenomenon that higher fuel moistures would lower the
combustion efficiency, shorten flaming phase and introduce prolonged smoldering combustion (Chen et al., 2010). The
relative proportion of flaming versus smoldering phases can vary considerably as a function of fuel moistures. Similar MCE
was also observed among wheat burning experiments with different levels of moisture contents (Table S1). This was because
that MCE is not only influenced by biofuel moisture contents but also the variations of biofuel structures (e.g. size), burning
temperatures or ambient conditions (Chen and Bond, 2010; Lu et al., 2009; Sanchis et al., 2014). We cannot completely
exclude the differences of other factors between each parallel experiment, which was the reason for repeating each condition
for three times in our experiment (Table S1).
Figure 1 displays variations of the monitored parameters (CO, $CO_2$, $\Delta CO/\Delta CO_2$ and fire temperature) during two
selected burning experiments (low-moisture biomass burning with MCE=0.83, and high-moisture biomass burning with
MCE=0.68). Different burning conditions dominate at different periods of a fire and the length of each period varied by
experiments (Figure 1). Actually, flaming and smoldering phases occur simultaneously during a fire and the proportions of
different combustion types vary over time (Akagi et al., 2011; Andreae and Merlet, 2001). For example, the initial period of
low-moisture biomass burning experiment (Figure 1a) is dominated by flaming, wherein $CO_2$ increased rapidly to the highest
level and $\Delta CO/\Delta CO_2$ ratios were lower (MCE was higher) compared with the smoldering-dominated period. The fire
temperatures were very high during this initially high-efficiency burning period. During the later period, smoldering
dominated the burning conditions. The burning efficiency and fire temperatures decreased during this period, and
$\Delta CO/\Delta CO_2$ ratios were higher than the first period. Previous ground-based and aircraft measurements of wildfire emissions
also observed gradually decreased combustion efficiency of a fire over time (Collier et al., 2016). For the high-moisture
biomass burning, smoldering combustion dominated the fire types during the whole period (Figure 1b). In higher-moisture
fuel burns, some energy released from the combustion is first used to dry up the higher moistures of the biofuels, thus the fire
temperatures and burning efficiency were lower than those of the low-moisture biomass burning.
**3.2 Emission factors of carbonaceous aerosols**
The average emission factors of $PM_{2.5}$, OC and EC were 9.3±3.4, 4.6±1.9 and 0.21±0.07 g/kg for corn burning and
8.7±5.0, 3.9±2.8 and 0.22±0.05 g/kg for wheat burning (Figure 2). The measured emission factors in this study fall within
the range of previous straw burning experiments (4.7-12.9, 1.2-8.9, 0.17-1.2 g/kg for $PM_{2.5}$, OC and EC, respectively)(Akagi
et al., 2011; Hays et al., 2005; Li et al., 2007). The estimated EFs from crop residue burning were generally lower than wood
or forest burning emissions (Akagi et al., 2011; Aurell and Gullett, 2013; Jen et al., 2019). However, open crop residue
burning in the field could result in severe air pollution during harvest season, especially in agriculture areas in China and
South Asia (IARI, 2012; Li et al., 2014; Streets et al., 2003a; Venkataraman et al., 2006). This type of biomass burning
cannot be neglected in these regions.
Organic matter (OM), calculated by multiplying OC by 1.3 (Li et al., 2007), was the dominant component of straw
burning aerosols, which accounted for ~64% and ~55% of the $PM_{2.5}$ emitted from corn and wheat burning (Figure 2).
Around 57% and 68% of the OC from corn and wheat burning are water soluble, and $HULIS_C$ represent 53% and 46% of the
WSOC. Though the mass contributions of WISOC were lower than WSOC in straw burning aerosols (Figure 2), the WISOC
fractions cannot be neglected, especially for considering the light absorption properties of BBOA (see section 3.4). Previous
studies also suggested a large portion of WISOC in ambient aerosols, which are important contributor of light-absorbing BrC
(Cheng et al., 2016b; Cheng et al., 2017).
The average EFs of water-soluble acetic acid, formic acid, succinic acid and oxalic acid were respectively 13.3±13.9,
4.1±3.3, 8.8±10.6, 2.2±1.1 mg/kg for corn burning and 13.0±14.5, 4.7±5.3, 9.9±13.5, 3.1±1.9 mg/kg for wheat burning
(Figure 2). Propionic acid and methanesulfonic acid in most samples were below the instrument detection limits in this study,
and their emissions were not taken into consideration in the following discussion. The quantified water-soluble
low-molecular-weight acids averagely accounted for 0.84% (0.16%-1.6%) and 0.88% (0.24%-1.8%) of the water-soluble
OM (WSOM) emitted from corn and wheat burning. Previous study has suggested that low molecular weight organic acids
represented an important fraction of WSOC in BBOA, and oxalic acid was a dominant short dicarboxylic (C2-C6) acids
(Falkovich et al., 2005). The estimated emission factors of acetic acid and formic acid in this work were lower than those
emitted from eucalypt forest fires, which were reported 17 and 26 mg/kg for flaming combustion, and 104 and 94 mg/kg for
smoldering combustion based on ground-based field measurements (Reisen et al., 2018). The difference could be attributed
to different biofuels, burning conditions as well as conducted experimental methods.
Figure 2 compares the emission factors of $PM_{2.5}$, carbonaceous aerosols and low molecular weight organic acids from
straw residue burning under different levels of moisture contents. The EFs of fine particles and organic carbonaceous
aerosols from high-moisture biomass burning were obviously higher than those from low-moisture biomass burning.
Substantial particulate carbonaceous aerosols could be generated from burning of higher-moisture biofuels, which is mainly
associated with the prolonged smoldering phases and less efficient combustions (Figure 1, Table S1). Similar variation trends
were also reported in previous biomass burning studies (Chen et al., 2010; Sanchis et al., 2014). Different levels of biofuel
moisture contents will actually influence the burning conditions, and thus impact the emission levels and compositions of
particulate matters.
**3.3 Influence of combustion efficiency on emission factors**
As shown in Figure 3, the emission factors of $PM_{2.5}$ and organic carbonaceous components increased with decreasing
MCE. Particle emissions were obviously enhanced under less efficient burning conditions. The emission factors of $PM_{2.5}$,
OC, WSOC and $HULIS_C$ from the most smoldering combustion experiment were about 3.4, 4.3, 3.8 and 2.8 times of those
from the most flaming combustion condition, regardless of the biomass types. The emissions of low molecular weight
organic acids also follow the similar variation trends with combustion efficiency as those of OC or WSOC emission factors
(Figure S3). These trends are generally in agreement with previous studies (Dhammapala et al., 2006; Holder et al., 2016;
Jen et al., 2019; Reisen et al., 2018; Wang et al., 2013). Under the same burning conditions, the emission factors of particles
or organic aerosols from corn burning were slightly higher than those from wheat burning (Figure 3). This was mainly due to
the different pyrolysis temperatures and combustion efficiency of different biofuels, which would influence the burning
processes (Khan et al., 2009; Zanatta et al., 2016). Our results suggested that the influence of varied burning conditions or
combustion efficiency on particle emissions could surpass the differences between the two types of straw residue burning
measured in this study (Figure 3). Thus, the burning efficiency or conditions should be taken into consideration when
simulate or estimate the influence of biomass burning emissions in future models.
Different from organic compounds, the emission factors of EC under different combustion efficiency remain relatively
consistent (Figure 3e). Holder et al. (2016) summarized the results from lab and field studies, and also found that the black
carbon emission factors from different studies are relatively constant, despite the differences in plume dilution or
measurement methods (Holder et al., 2016). Some studies, however, reported an increasing trend in EC or BC emission
levels with the increasing of combustion efficiency in wildfires or forest burns in U.S. (Aurell and Gullett, 2013; Jen et al.,
2019). As the conducted experiments were mostly dominated by smoldering combustions (MCE=0.68-0.88) in this study, we
cannot exclude the possibility that the EC emissions may be higher under flaming-dominated combustions (e.g. MCE>0.9).
Though the EC emission factors did not show obvious variation trends as a function of MCE, a positive correlation between
EC/(OC+EC) ratios and combustion efficiency was observed (Figure 3g). Due to the obvious dependence of EC/OC or
EC/(OC+EC) ratios on burning efficiency, these ratios could be employed to indicate different burning conditions when the
emitted CO and $CO_2$ data are not available, which have been used in previous studies (Xie et al., 2018; Xie et al., 2019).
To further investigate the influence of burning conditions on the chemical compositions of biomass burning aerosols,
mass ratios of WSOC/OC, $HULIS_C$/OC, $K^+$/OC and $Cl^-$/OC as a function of burning efficiency are plotted in Figure 4. The
WSOC/OC and $HULIS_C$/OC mass ratios ranged from 0.52-0.78 and 0.16-0.54 among different burning experiments. The
$HULIS_C$/OC ratios were comparable to those (0.26-0.44, with an average of 0.34) reported in field or controlled chamber
combustion experiments (Lin et al., 2010). We did not observe obvious variation trends of WSOC/OC or $HULIS_C$/OC ratios
with MCE (Figure 4), which indicated relative constant BBOA chemical compositions under different combustion conditions.
However, the $K^+$/OC and $Cl^-$/OC ratios showed consistent variation trends under different MCE conditions, which increased
from <0.1 under the more smoldering condition to >0.5 under the more flaming condition for $K^+$/OC, and from 0.05 to >0.5
for $Cl^-$/OC (Figure 4). The highest $K^+$/OC (0.64) and $Cl^-$/OC (0.61) ratios were observed in a low-moisture wheat burning
experiment with a MCE of 0.79. This is because the K and Cl emissions from combustion are highly affected by fire
temperatures and burning conditions. Lab-controlled experiments suggested that the proportions of released potassium and
chlorine from the biomass fuels increase with the applied combustion temperatures (Jensen et al., 2000; Knudsen et al.,
2004). The flaming combustion (with higher MCE) was observed much higher fire temperatures than the smoldering
combustion (with lower MCE) (Figure 1). Though the emission levels of particles or organic aerosols decreased during
higher efficiency burning (Figure 3), elevated proportions of potassium and chlorine were released into smokes during the
flaming combustion phase under this condition (Figure 4b). The two wheat burning experiments (moisture content=7%) with
higher $K^+$/OC and $Cl^-$/OC ratios (>0.5) than others were related to the higher combustion temperatures during the initial
flaming periods of the burning experiments (Figure S4). Potassium ion is a commonly used tracer to indicate the biomass
burning emissions. However, our results revealed that $K^+$ cannot correctly indicate the emission levels of biomass burning
aerosols under obviously different burning conditions, which may lead to large uncertainty in estimating burning emissions if
without considering the combustion conditions.
**3.4 Light absorption of BBOA**
The light absorption of straw burning organic aerosols decreased sharply from near-UV to visible wavelengths (Figure
5), indicating their properties as biomass burning-generated BrC. The absorption of WISOC, WSOC and HULIS$_C$ at 300 nm
was as high as 4.5, 15.2 and 11.2 times of those at 400 nm for corn burning emissions, and 4.8, 9.2 and 10.6 times for wheat
burning emissions. The wavelength dependence property of BBOA light absorption was described by AAE derived from the
absorption in the range of 300-450 nm. The AAE of WISOC, WSOC and HULIS$_C$ were respectively 5.8-5.9, 8.6-11.3,
8.9-10.2 for corn burning aerosols and 5.7-6.0, 8.1-9.0, 9.0-10.5 for wheat burning aerosols, and the averaged values were
also shown in Figure 5. The water-soluble BBOA fractions (WSOC and HULIS) showed stronger wavelength dependence
than the water insoluble fractions. The estimated AAE values of straw burning organic aerosols in this study are comparable
to those of BBOA (5.3-8.1) and biomass burning-influenced atmospheric aerosols (5.2-9.4) reported in previous studies
(Hecobian et al., 2010; Hoffer et al., 2006; Wu et al., 2018; Wu et al., 2019; Xie et al., 2017; Xie et al., 2019; Zhu et al.,
2018). The strong light absorption of biomass burning-generated BrC in near-UV range would lead to an increase in aerosol
light absorption and radiative forcing efficiency (Chakrabarty et al., 2010).
The WISOC was the most important light-absorption fraction among straw burning organic aerosols, which contributed
61%−84% and 57%−72% of the light absorption (@300-400 nm) by extracted BrC emitted from corn and wheat burning
(Figure 5). In the wavelength range of 300-400 nm, HULIS$_C$ and other high-polarity WSOC (WSOC-h=WSOC-HULIS$_C$)
respectively contribute to 16%-28% and 1%-10% of the total BBOA absorption for corn burning, and 17%-29% and 12%-15%
for wheat burning. Though the mass contribution of WISOC was lower than WSOC (Figure 2), the light absorption of
WISOC surpassed WSOC due to the higher light absorption capability of water-insoluble BBOA, indicated by the higher
MAE of WISOC (Figure 6). Meanwhile, the light absorption of water-soluble BBOA among near-UV ranges cannot be
neglected due to their sharper increase of absorption towards shorter wavelength compared with WISOC (Figure 5). The
light absorption contribution of WSOC to extracted BrC increased substantially from 16%-28% at 400 nm to 39%-43% at
300 nm. Among the water-soluble BBOA, HULIS were the major contributors of light absorption, which occupied 74% and
68% of the WSOC absorption at 300 nm for corn and wheat burning emissions, respectively. This was due to the higher light
absorption capability of HULIS than other high-polarity WSOC fractions (Figure 6), though their mass contributions were
comparable in straw burning aerosols (Figure 2).

305        The light absorption capabilities of different BBOA fractions are compared in Figure 6. The estimated $MAE_{365}$ values of

straw burning-generated BrC in this study are comparable to those reported in previous studies (Fan et al., 2018; Xie et al.,
2017). The MAE of WISOC are higher than water-soluble BBOA (WSOC and HULIS) among the measured wavelength
ranges for both corn and wheat burning aerosols. The $MAE_{300}$ of WISOC was 1.6 and 1.7 times of WSOC emitted from corn
and wheat burning, and comparable to those of HULIS (Figure 6). Due to the slower decrease of WISOC absorption towards
visible wavelengths than the water-soluble fractions (Figure 5), the $MAE_{365}$ of WISOC was as high as 2.5 and 2.2 times of
WSOC from corn and wheat burning emissions, and 1.7 and 1.6 times of HULIS. Though the mass contribution of WISOC
among BBOA could be smaller than WSOC, their contribution to light absorption cannot be neglected due to the higher
MAE of water insoluble BBOA. The solar energy absorbed by biomass burning-emitted WISOC relative to EC (25%)
among the wavelength range of 300-700 nm was higher than those of WSOC (10%) or HULIS (4%). The light absorption of
BBOA would be largely underestimated if only considering the water soluble fractions. Previous studies also reported a large
proportion of WISOC absorption in BBOA and ambient aerosols (Cheng et al., 2016b; Cheng et al., 2017; Park et al., 2018;
Sengupta et al., 2018).

318        Figure 7 clearly shows the dependence of BBOA absorption coefficient ($Abs_{365}$) on burning conditions. Higher $Abs_{365}$

of biomass burning-generated BrC were observed under less efficient burning conditions for both corn and wheat burning
experiments. This is mainly due to the elevated BBOA emission factors as MCE decreases (Figure 3). We did not observe
obvious dependence of $MAE_{365}$ on the combustion efficiency for either water-soluble fractions or WISOC (Figure 7d-f).
Previous lab and field studies suggested that the optical properties of biomass burning aerosols are more dependent on
burning conditions other than fuel types (Liu et al., 2014; Xie et al., 2017). The $MAE_{365}$ of BBOA emitted from flaming
combustion were reported higher than those from smoldering combustion based on lab-controlled burning experiments (Xie
et al., 2019). Another lab experiment also suggested the dependence of $MAE_{365}$ of methanol-extracted BBOA on burning
conditions, while the variation trends are different regarding different fuel types or sampling methods among different
experiments (Xie et al., 2017). It is noted that limited sample population was selected to conduct the light absorption
measurements and smoldering dominated the burning conditions in this study, which could be the reasons that we did not
observe an obvious dependence of MAE on combustion conditions. More lab experiments, involving larger numbers of
experiments and more variable burning conditions, are required to address the influence of combustion efficiency on light
absorption capability of biomass burning-emitted carbonaceous aerosols in future studies.

## 4 Conclusions

The emission factors of $PM_{2.5}$, OC and EC were 9.3, 4.6 and 0.21 g/kg for corn burning and 8.7, 3.9 and 0.22 g/kg for
wheat burning, generally lower than wood or forest burning emissions. Around 57% and 68% of the OC emitted from corn
and wheat burning are WSOC, among which $HULIS_C$ represent 53% and 46% of the WSOC mass concentrations. Though
the mass contribution of WISOC was lower than WSOC, the light absorption contribution of WISOC (57%−84% @300-400
nm) surpassed WSOC due to the higher MAE of WISOC. The BBOA light absorption would be largely underestimated if
only considering the water soluble fractions. Meanwhile, the light absorption of WSOC among near-UV ranges, occupying
39%-43% of extracted OC absorption at 300 nm, cannot be neglected due to their sharper increase of absorption towards
shorter wavelength compared with WISOC. HULIS were the major light absorption contributors among WSOC, and their
light absorption capability was higher than other high-polarity WSOC components.
The emission levels, compositions and light absorption of BBOA were influenced by the burning conditions. The
combustion conditions varied from different burning experiments, with the MCE ranging from 0.68 to 0.88. The emission
factors of $PM_{2.5}$ and organic carbonaceous aerosols were obviously enhanced under less efficient burning conditions (lower
MCE). The emission factors of $PM_{2.5}$, OC, WSOC and $HULIS_C$ from the most smoldering combustion experiment were
about 3.4, 4.3, 3.8 and 2.8 times of those from the most flaming combustion condition, regardless of the biofuel types
employed in this study. The emission factors of $PM_{2.5}$ and carbonaceous aerosols from high-moisture straw burning were
obviously elevated compared with those from low-moisture straw burning experiments. This is mainly due to the prolonged
smoldering and incomplete combustion period during high-moisture biomass burning.
The EC/(EC+OC) ratios showed a positive correlation with MCE, though EC emission factors remain relative constant
under different combustion conditions. Thus, it is reasonable to employ EC/OC or EC/(EC+OC) ratios as an indicator of
biomass burning conditions. The mass ratios of WSOC/OC or $HULIS_C$/OC did not display obvious variation trends under
different combustion efficiency. However, the $K^+$/OC and $Cl^-$/OC ratios showed continuous increasing trends during higher
efficiency burning, from <0.1 under the more smoldering condition to >0.5 under the more flaming condition for $K^+$/OC, and
from 0.05 to >0.5 for $Cl^-$/OC ratios. This is mainly attributed to the elevated proportions of released potassium and chlorine
from biofuels under the higher fire temperatures during flaming combustions. Our results indicate that potassium ion, as a

commonly used biomass burning tracer, may lead to large uncertainty in estimating biomass burning emission levels without considering the combustion conditions.

Higher absorption coefficient ($Abs_{365}$) of straw burning-generated BrC, including WSOC, HULIS and WISOC, were observed under less efficient burning conditions for both corn and wheat burning. This is mainly attributed to the higher BBOA emission factors as MCE decreases. Our results suggested that the influence of varied combustion efficiency on the emission levels of BBOA could surpass the differences between biofuel types. Thus, the burning efficiency or combustion conditions should be taken into consideration when estimate the influence of biomass burning.

*Data availability.* The data presented in this article are available from the authors upon request (minhu@pku.edu.cn).

The Supplement related to this article is available online

*Author contributions.* MH, ZW, XH, and LH organized the project. YW conducted the simulation experiments. YW, NX and YQ analyzed the samples. YW wrote the manuscript with input from all co-authors. All authors contributed to discussing the results and commenting on the manuscript.

*Competing interests.* The authors declare that they have no conflict of interest.

*Acknowledgements.* This study was supported by National Natural Science Foundation of China (91844301, 91544214), and the project funded by China Postdoctoral Science Foundation (2019M650354). We also thank Dr. Song Guo for his helpful suggestions on this study.

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

**Figures**

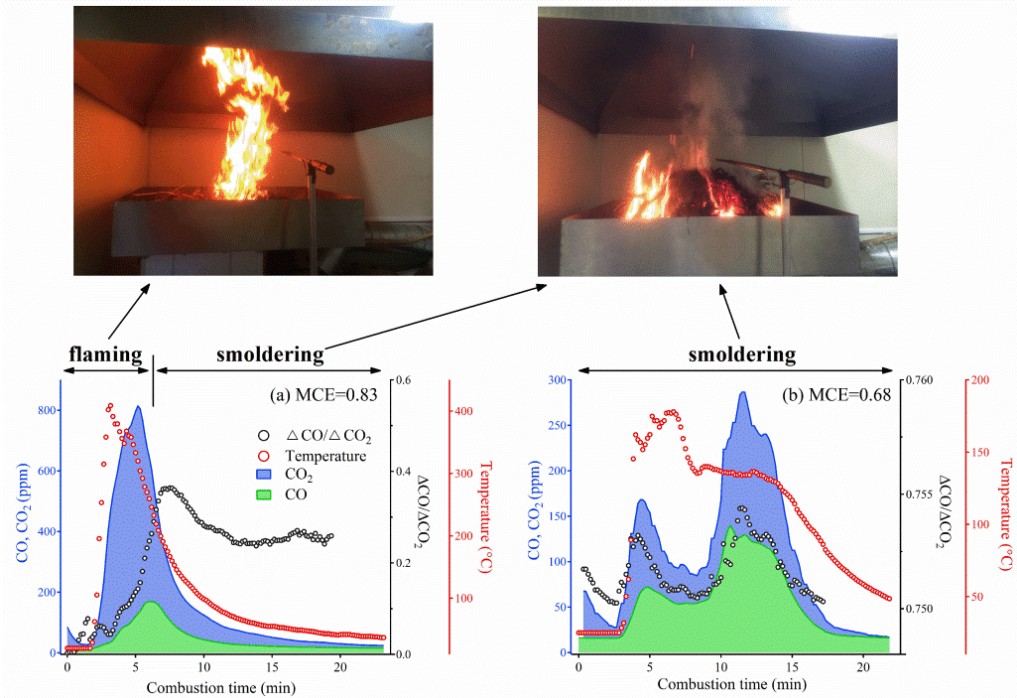


Figure 1 Variations of measured CO, $CO_2$ concentrations, $\Delta CO/\Delta CO_2$, fire temperatures and burning conditions during two
selected experiments, with an averaged MCE value of (a) 0.83 and (b) 0.68.

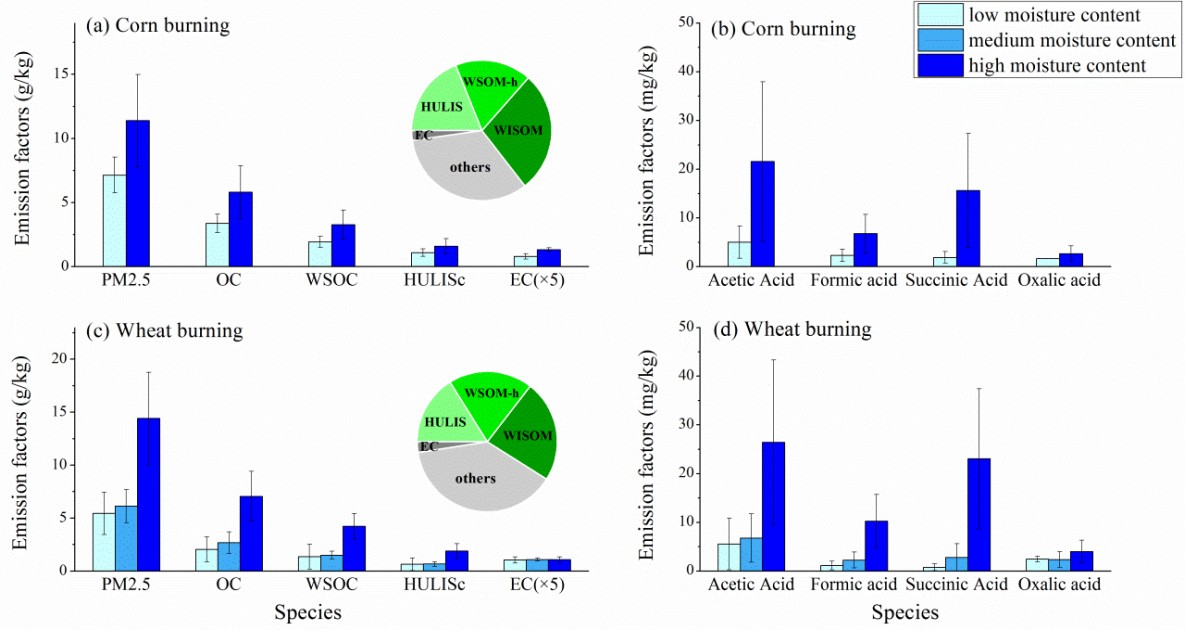


Figure 2 Emission factors of $PM_{2.5}$, OC, WSOC, $HULIS_C$ and EC from (a) corn burning and (c) wheat burning, and emission
factors of low molecular weight organic acids (acetic acid, formic acid, succinic acid, and oxalic acid) from (b) corn burning
and (d) wheat burning. The EC emission factors are represented by 5×EC due to the low values. The pie charts in panels (a)
and (c) represent the contribution of major carbonaceous aerosols among $PM_{2.5}$. The high-polarity WSOM (WSOM-h) is
calculated by subtracting HULIS from WSOM. Different moisture content levels correspond to those shown in Table S1.


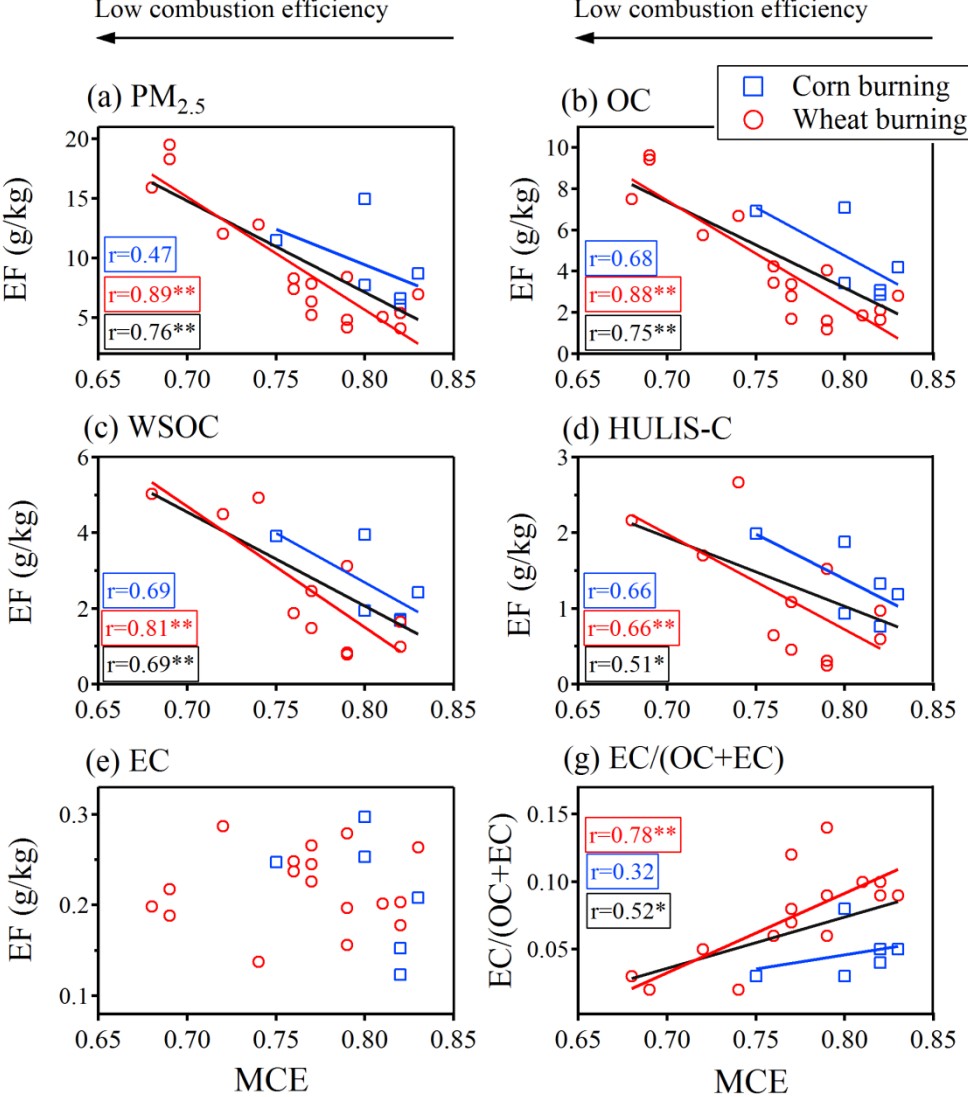


Figure 3 Emission factors of PM$_{2.5}$, carbonaceous aerosols (OC, WSOC, HULIS$_C$ and EC) and EC/(OC+EC) ratios as a
function of modified combustion efficiency (MCE). Corn and wheat burning emissions are denoted by red and blue colors,
respectively. The r values in each panel are the Pearson correlations between emission factors and MCE for corn (blue),
wheat (red) and the overall (black) burning experiments. The ** or * following the r value indicates the correlation is
significant at the 0.01 level or 0.05 level (2-tailed).


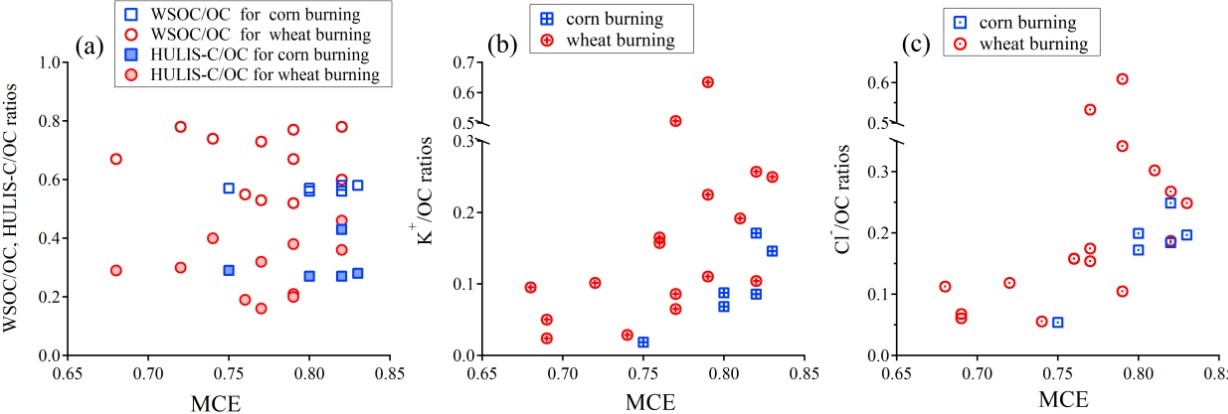

Figure 4 Variations of (a) WSOC/OC and HULIS$_C$/OC ratios, (b) K$^+$/OC, and (c) Cl$^-$/OC ratios as a function of modified combustion efficiency (MCE) for corn and wheat burning experiments. Corn and wheat burning emissions are denoted by red and blue colors, respectively.

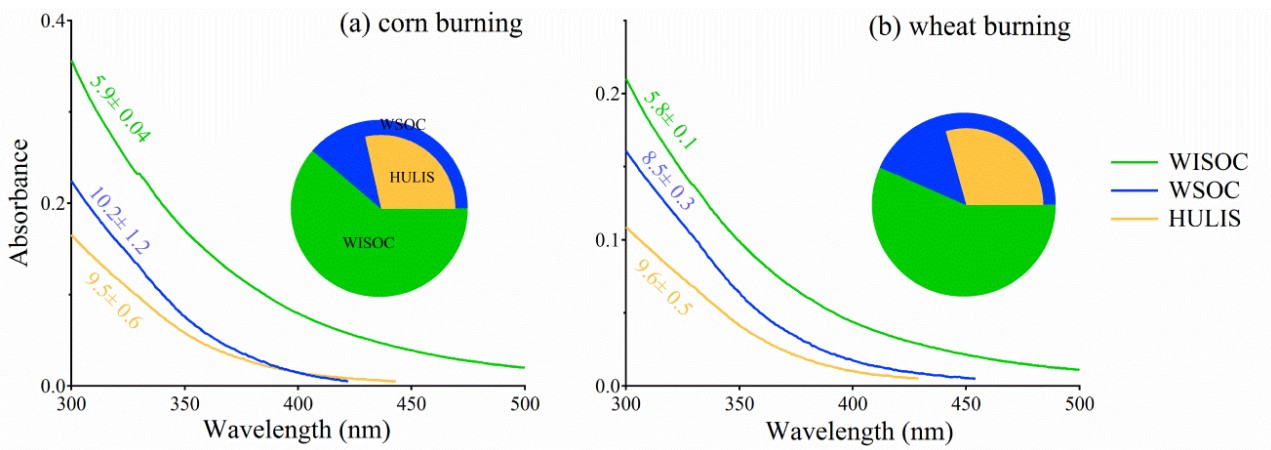

Figure 5 UV-vis spectra of carbonaceous aerosol solutions, including WSOC, HULIS$_C$ and WISOC, from (a) corn and (b) wheat burning experiments. The pie chart in each panel is the absorption contribution of different BBOA fractions at 300 nm. The number represents the average AAE of each BBOA fraction derived from the absorption in the wavelength range of 300-450 nm.

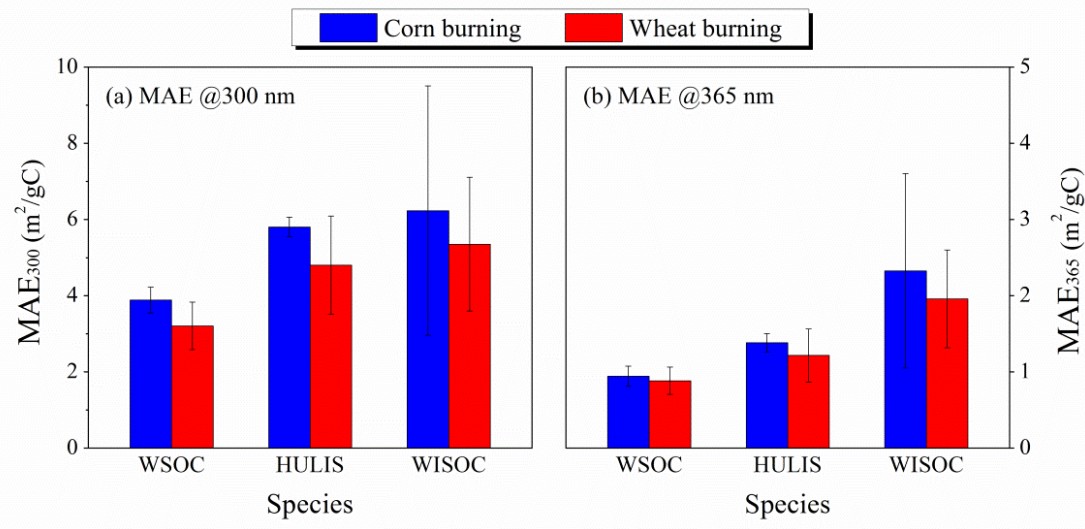

Figure 6 Mass absorption efficiency (MAE) of different organic carbonaceous aerosols, including WSOC, HULIS and

WISOC emitted from corn and wheat burning.

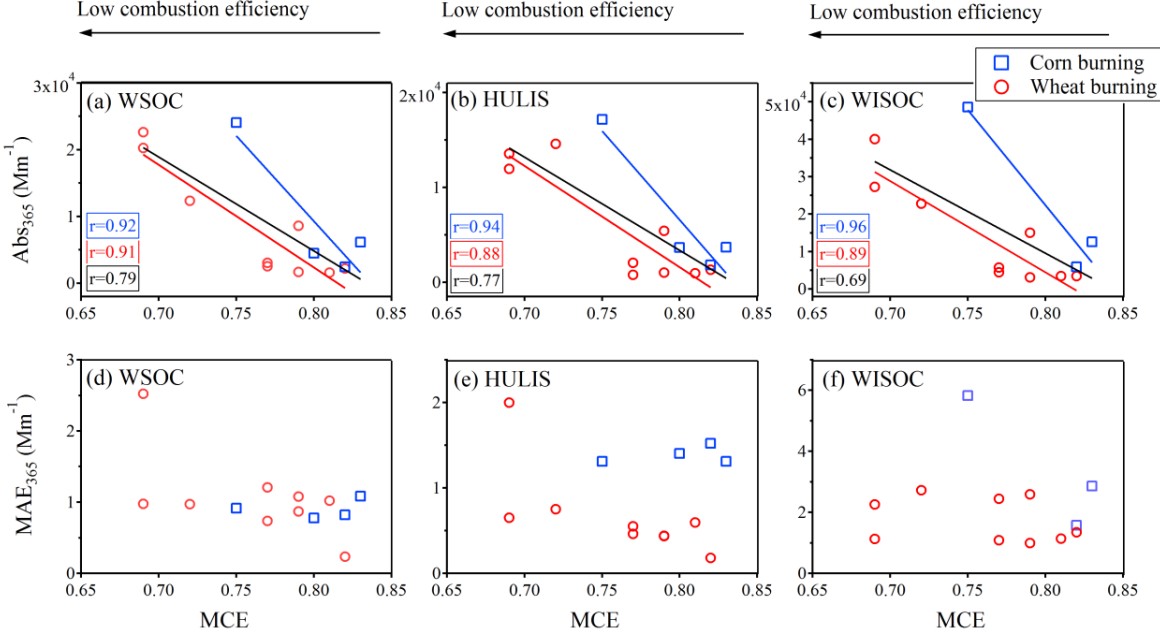

Figure 7 (a-c) Light absorption coefficients (Abs$_{365}$) and (d-f) mass absorption efficiency (MAE$_{365}$) of WSOC, HULIS$_C$ and

WISOC at 365 nm as a function of combustion efficiency. Corn and wheat burning emissions are denoted by red and blue

colors, respectively. The r values in panels (a-c) are the correlation coefficients for corn (blue), wheat (red) and overall (black)

burning experiments.