# Peer review of "Chemical composition and light absorption of carbonaceous aerosols emitted from crop residue burning: Influence of combustion efficiency"

_Atmospheric Chemistry and Physics, 2020_

## Referee Comment (RC1) · Anonymous Referee #1 · 29 Aug 2020

This study investigated the emission factors (EFs) and light absorption of carbonaceous components, including water-soluble organic carbon (WSOC), humic-like substances (HULIS), and water-insoluble organic carbon (WISOC), from burning crop residues (wheat and corn). Also, the influences of biofuel moisture and burning conditions on EFs and brown carbon (BrC) absorption were analyzed by using data of modified combustion efficiency (MCE). Although a clear dependence of EFs on MCE values was illustrated, the influence of burning conditions on biomass burning (BB) BrC absorption can hardly be observed or validated. This might be due to the limit in sample number, smoldering combustion conditions (MCE = 0.68 – 0.88), or uncertainties in the calculation of mass absorption efficiency ($MAE_\lambda$). Before the acceptance to publication, the following issues should be addressed.

1. Page 5, line 137. Why total OC was used to represent the concentration of extracted OC?

In Page 127, it was stated that "total OC was analyzed by a thermal/optical carbon analyzer (Subset Laboratory)".

These two "total OC" should be different. The first is used to calculate EFs of OC, while the second is derived for $MAE_\lambda$ calculation.

Furthermore, the authors can perform better estimation on extracted OC (or WISOC) mass by measuring residue OC on filter samples after solvent extractions. Then the calculation of $MAE_\lambda$ of WISOC will be less uncertain.

Typically, the residue OC would account for ~10% of the total, and WSOC contributed more than 50% of total OC in this work. Then the inter-sample variability of residue OC will lead to substantial uncertainty on the estimation of WISOC mass and absorption.

2. The dependence of EFs on burn conditions was well illustrated in Figures 3 and 4. But Figure 7d-f did not show any influence of burn conditions on light absorption. Figure 7a-c and Figure 3b-d tell the same thing——smoldering combustion has higher EFs of carbonaceous aerosols.

Page 11, lines 296-297, "Furthermore, the $MAE_{365}$ of WSOC and HULIS emitted from straw burning were slightly higher under less efficient burning conditions (Figures 7d, 7e)"

In previous studies, MAE values tend to be greater under more flaming conditions or higher burning temperatures. The observation results reported here seems not reasonable.

Due to the sample number limit and small variability in $MAE_{365}$ for most observations, the light absorption of BB BrC did not show any dependence on burn conditions.

Page 12, lines 339-342, the final conclusion "Our results suggested that the influence of varied combustion efficiency on the emission levels and light absorption of BBOA

could surpass the differences between biofuel types. Thus, the burning efficiency or combustion conditions should be taken into consideration when estimate the influence of biomass burning." was not fully supported by the experiments results.

---

## Referee Comment (RC2) · Anonymous Referee #2 · 7 Sep 2020

The paper by Wang et al. summarizes results on aerosol emission factors and optical properties in burning of agricultural residues (wheat and corn straw) under different burn conditions. They determine the emission factors of PM2.5, EC, OC, and different components of OC (water soluble, including HULIS and low-molecular weight oxygenated molecules, and the insoluble fraction) and also determine the wavelength-dependent absorbance, mass absorption efficiency, and Angstrom Exponent of Absorption. They highlight that the EFs of all species except EC was higher at the lower combustion efficiency values (estimated by measurements of CO and CO2) and that the WISOC had the largest contribution to the measured absorbance; however, wavelength dependence of absorption was strongest for the WSOC and HULIS. The results

are interesting to the community and the paper fits the scope of ACP. The paper is overall well written although some parts benefit from some editing (I suggest below). I'd like the authors to clarify the points I highlight below before the paper is accepted for publication:

Technical points: L76-77: I think this statement underestimates all the studies that have been carried out in the Fire lab in Missoula, that characterize influence of combustion efficiency on aerosol optical properties. I can imagine that for agricultural residue burning, the studies are limited, so a more accurate statement should be included here.

L110-111: Do authors mean that fuels were weighed before and after drying? If so please add this detail.

Table S1: There doesn't seem to be a consistent picture between MCE and the moisture content. For example, MCE ∼0.77 was observed at all different moisture content values of the wheat. Please explain the reason for this variability. Because of this lack of obvious trend, I would not mention this in the conclusions either (L318-319)

Eqn of Abs(l): why is absorbance at 700 nm subtracted from the absorbance at the wavelength of interest? Why should this be a relative absorbance? Shouldn't the absorbance at a specific wavelength be corrected for the background absorbance at the same wavelength while sampling only pure water?

L178-180: The average EFs of corn are higher, but still considering the variabilities that were observed for both fuels, the difference isn't significant and beyond the observed variabilities.

Fig. 3g: why not showing all the fits as in the other panels? Also, are the fits a double-sided regression line, considering the uncertainties in the x and y values?

L233: what precluded the possibility of having burns with MCE>0.9 that's more representative of flaming conditions? I think some discussion should be provided. Also it would be valuable to mention what the expected MCE in real world burns of agricultural

residues are so readers get an idea of how applicable the results are and what values are most meaningful to be used in models.

Figure 4. There are some wheat burns for which the K+/OC and Cl-/OC ratios are highly variable; are all the burns from the same batch of fuel? Could this variability be explained by variable K and Cl content of the fuel itself?

L290-291: Since this paper has reported on MAE as well as EF of the different components of OC, it will be very valuable to combine the two results and present the EF of absorption to be able to more directly compare radiative impacts of WISOC, WSOC, and HULIS.

L297-298, 338-339: I disagree; there are really two points that might be considered as outliers and without those, the MAE(365) vs MCE looks pretty flat. I suggest removing this statement.

Suggested Edits: L 33: remove observed in ".weere also observed higher under…" L39 and 334: remove if in "if without considering the burning conditions…" L 61: Add "…was reported to be higher for more …"

L112: consider changing "weighted" to "weighed"

L118: include the volumetric unit for both 10 and 5 units of water and methanol, respectively.

L121: why did you use a smaller size filter for the WISOC fraction?

L129-L130: consider changing "minus…" to "difference between total OC and WSOC." Eqns. Consider adding equation numbers

L174-175: I think I know what the authors try to say (in higher moisture fuel burns, some energy is first used to dry up the fuel and so the temperature is lower); however, as written the sentence is confusing. Consider rephrasing it.

L186 and 191: change negligible to "neglected"

L200, 315: change "dominated" to "dominant"

L234-235: rephrase the beginning of the sentence; the structure is not correct

L237: data "are"...

L247: remove "that"

L272: consider changing "occupy" to "contribute to"

L271-273: the % contributions are for 300 nm and 400 nm, respectively? It's unclear when a range of 300-400nm is mentioned. Please clarify.

L296, 338: change "as the decreasing of MCE" to "... as MCE decreases..."

---

## Author Comment (AC1) · 30 Sep 2020

Dear editor and reviewers,

We appreciate all your detail and valuable suggestions on our manuscript (acp-2020-676). We have carefully considered the comments and revised the manuscript accordingly. Please see the point-by-point response below and changes are marked blue in the revised manuscript.

Thanks for your kind help.

Best regards,

Min Hu

**Point-by-point response to review comments**

**Referee #1**

*This study investigated the emission factors (EFs) and light absorption of carbonaceous components, including water-soluble organic carbon (WSOC), humic-like substances (HULIS), and water-insoluble organic carbon (WISOC), from burning crop residues (wheat and corn). Also, the influences of biofuel moisture and burning conditions on EFs and brown carbon (BrC) absorption were analyzed by using data of modified combustion efficiency (MCE). Although a clear dependence of EFs on MCE values was illustrated, the influence of burning conditions on biomass burning (BB) BrC absorption can hardly be observed or validated. This might be due to the limit in sample number, smoldering combustion conditions (MCE = 0.68 – 0.88), or uncertainties in the calculation of mass absorption efficiency ($MAE_\lambda$). Before the acceptance to publication, the following issues should be addressed.*

> **Response**: Thanks for your valuable comments on our manuscript and pointing out the deficiency in discussing the influence of burning conditions on biomass burning BrC absorption. We have now carefully revised the manuscript and addressed the following comments.

*1. Page 5, line 137. Why total OC was used to represent the concentration of extracted OC?*

*In Page 127, it was stated that "total OC was analyzed by a thermal/optical carbon analyzer (Subset Laboratory)".*

*These two "total OC" should be different. The first is used to calculate EFs of OC, while the second is derived for $MAE_\lambda$ calculation.*

*Furthermore, the authors can perform better estimation on extracted OC (or WISOC) mass by measuring residue OC on filter samples after solvent extractions. Then the calculation of $MAE_\lambda$ of WISOC will be less uncertain.*

*Typically, the residue OC would account for ~10% of the total, and WSOC contributed more than 50% of total OC in this work. Then the inter-sample variability of residue OC will lead to substantial uncertainty on the estimation of WISOC mass and absorption.*

> **Response**: Thanks for pointing out the uncertainties in estimating "extracted OC" concentrations. Yes, we used the total OC analyzed by the thermal/optical carbon analyzer to calculate the EFs and also to represent the "total extracted OC" to derive the $MAE_\lambda$. We agree with the referee that this may lead to an underestimation of MAE of WISOC, which we mentioned in lines 143-144.
>
> The OC concentration measured by the thermal/optical carbon analyzer has been widely employed to estimate the absorption capability of OC or WISOC extracted by methanol in previous studies (Liu, 2014; Zhu et al., 2018). Chen and Bond (2010) suggested that more than 92% of OC emitted from biomass pyrolysis could be extracted by methanol. Xie et al. (2019) suggested that 93.6%-99.7% of biomass burning-generated OC could be extracted by methanol (added in lines 144-146). Thus, the residue OC was relatively small compared with the

methanol-extracted OC, and the difference between the total WISOC (estimated by the difference between thermal/optical carbon analyzer measured OC and WSOC) and methanol-extracted WISOC were relatively small and not taken into consideration in this study.

We agree with the referee that it is very important to estimate the residue OC and make an accurate calculation of WISOC absorption. However, during the extraction procedures in this study, we first cut up the filter samples and then extracted them by water and then methanol. The thermal/optical carbon analyzer quantified OC on certain size filter by converting the carbon species to methane, and measuring by a flame ionization detector. Thus, it's difficult for us to measure the residue OC due to the organic solvent absorbed on the extracted filters and the fragmentized filters after extraction. We will design experiments, such as those conducted in Chen and Bond (2010) and Xie et al. (2019), to estimate the residue OC concentrations and make a more accurate calculation of the MAE of WISOC in our future studies.

Lines 143-146:

"It is noted that the total OC was used to represent the concentration of total extracted OC, which may lead to an underestimation of MAE of WISOC. Previous studies suggested that 92%-99.7% of BBOA could be extracted by methanol (Chen and Bond, 2010; Xie et al., 2019), thus the residue OC un-extracted by methanol was relatively small compared with the extracted fraction."

References:

Chen, Y. and Bond, T. C.: Light absorption by organic carbon from wood combustion, *Atmos. Chem. Phys.*, 10, 1773-1787, 10.5194/acp-10-1773-2010, 2010.

Liu, J., Scheuer, E., Dibb, J., Ziemba, L. D., Thornhill, K. L., Anderson, B. E., Wisthaler, A., Mikoviny, T., Devi, J. J., Bergin, M., and Weber, R. J.: Brown carbon in the continental troposphere, *Geophys. Res. Lett.*, 41, 2191-2195, 10.1002/2013gl058976, 2014.

Xie, M., Chen, X., Hays, M. D., and Holder, A. L.: Composition and light absorption of N-containing aromatic compounds in organic aerosols from laboratory biomass burning, *Atmos Chem Phys*, 19, 2899-2915, 10.5194/acp-19-2899-2019, 2019.

Zhu, C. S., Cao, J. J., Huang, R. J., et al.: Light absorption properties of brown carbon over the southeastern Tibetan Plateau, *Sci. Total Environ.*, 625, 246-251, 10.1016/j.scitotenv.2017.12.183, 2018.

2. *The dependence of EFs on burn conditions was well illustrated in Figures 3 and 4. But Figure 7d-f did not show any influence of burn conditions on light absorption. Figure 7a-c and Figure 3b-d tell the same thing——smoldering combustion has higher EFs of carbonaceous aerosols.*

*Page 11, lines 296-297, "Furthermore, the MAE365 of WSOC and HULIS emitted from straw burning were slightly higher under less efficient burning conditions (Figures 7d, 7e)"*

*In previous studies, MAE values tend to be greater under more flaming conditions or higher burning temperatures. The observation results reported here seems not reasonable.*

*Due to the sample number limit and small variability in MAE365 for most observations, the light*

*absorption of BB BrC did not show any dependence on burn conditions.*

**Response**: We deleted the statement "*Furthermore, the MAE$_{365}$ of WSOC and HULIS emitted from straw burning were slightly higher under less efficient burning conditions.*" in the revised version and revised the sentence as in lines 320-321. We carefully checked the correlation between MAE$_{365}$ and MCE again, and found their correlations were not significant at the 0.01 level (2-tailed) for either WSOC or HULIS. The slight increasing trends of MAE$_{365}$ for WSOC and HULIS as the decreasing of MCE were really due to the two outliers in wheat burning experiments.

We agree with the referee that limited sample number conducted light absorption measurements and small variability in MAE$_{365}$ for most observations may be the reasons for lack of dependence of MAE on burning conditions. We added this reasons in the revised manuscript (lines 310-312). In this study, we intended to simulate the real combustion conditions of agriculture residue burning in the field. Smoldering-dominated conditions, with expected MCE<0.9 or even lower, have been widely observed during the burns of agricultural residues in the agricultural area in China (Figure R1) and India (Figure R2). Thus, the burning conditions were controlled to be dominated by smoldering (as shown in Figure 1, with MCE=0.68-0.88) in our experiments, and the small variation in burning conditions could be the reasons for small variability in MAE$_{365}$ for most observations. More lab-controlled burning experiments, involving larger numbers of experiments and more variable burning conditions, are required in our future studies to address the influence of combustion conditions on the BrC absorption (lines 327-331).

[Figure]

Figure R1 Intense straw burning in agriculture area (Anhui province, China) in China. (Wang, et al., 2017)

[Figure]

Figure R2 Post-harvest crop residue burning in northwest India. (IARI, 2012)

Lines 320-321:

"We did not observe obvious dependence of $MAE_{365}$ on the combustion efficiency for either water-soluble fractions or WISOC (Figure 7d-f)."

Lines 327-331:

"It is noted that limited sample population was selected to conduct the light absorption measurements and smoldering dominated the burning conditions in this study, which could be the reasons that we did not observe an obvious dependence of MAE on combustion conditions. More lab experiments, involving larger numbers of experiments and more variable burning conditions, are required to address the influence of combustion efficiency on light absorption capability of biomass burning-emitted carbonaceous aerosols in future studies."

References:

Crop Residues Management with Conservation Agriculture: Potential, Constraints and Policy Needs, edited by: Institute, I. A. R., India, 2012.

Wang, Y., Hu, M., Lin, P., et al.: Molecular characterization of nitrogen-containing organic compounds in humic-like substances emitted from straw residue burning, *Environ. Sci. Technol.*, 51, 5951-5961, 10.1021/acs.est.7b00248, 2017.

*Page 12, lines 339-342, the final conclusion "Our results suggested that the influence of varied combustion efficiency on the emission levels and light absorption of BBOA could surpass the differences between biofuel types. Thus, the burning efficiency or combustion conditions should be taken into consideration when estimate the influence of biomass burning." was not fully supported by the experiments results.*

**Response**: We agree with the referee that the influence of burning conditions on MAE of BBOA was not obvious in this study, as mentioned above. We revised this sentence as follows: "Our results suggested that the influence of varied combustion efficiency on the emission levels of BBOA could surpass the differences between biofuel types. (lines 361-362)". The emission factors of $PM_{2.5}$ or BBOA (OC, WSOC, HULIS) from more smoldering conditions were 2.8-4.3 times of those from more flaming conditions in the present study. While the differences between

wheat burning and corn burning under similar combustion conditions (or MCE) were not such obvious, as shown in Figures 3.

---

## Author Comment (AC2) · 30 Sep 2020

Dear editor and reviewers,

We appreciate all your detail and valuable suggestions on our manuscript (acp-2020-676). We have carefully considered the comments and revised the manuscript accordingly. Please see the point-by-point response below and changes are marked blue in the revised manuscript.

Thanks for your kind help.

Best regards,

Min Hu

**Point-by-point response to review comments**

**Referee #2**

*The paper by Wang et al. summarizes results on aerosol emission factors and optical properties in burning of agricultural residues (wheat and corn straw) under different burn conditions. They determine the emission factors of PM$_{2.5}$, EC, OC, and different components of OC (water soluble, including HULIS and low-molecular weight oxygenated molecules, and the insoluble fraction) and also determine the wavelength-dependent absorbance, mass absorption efficiency, and Angstrom Exponent of Absorption. They highlight that the EFs of all species except EC was higher at the lower combustion efficiency values (estimated by measurements of CO and CO2) and that the WISOC had the largest contribution to the measured absorbance; however, wavelength dependence of absorption was strongest for the WSOC and HULIS. The results are interesting to the community and the paper fits the scope of ACP. The paper is overall well written although some parts benefit from some editing (I suggest below). I'd like the authors to clarify the points I highlight below before the paper is accepted for publication:*

> **Response**: Thanks for your valuable comments on our manuscript. We have now carefully revised the manuscript and addressed the following points.

**Technical points:**

*L76-77: I think this statement underestimates all the studies that have been carried out in the Fire lab in Missoula, that characterize influence of combustion efficiency on aerosol optical properties. I can imagine that for agricultural residue burning, the studies are limited, so a more accurate statement should be included here.*

> **Response**: Thanks for the kind reminding. We added the related studies from the Fire lab in Missoula in lines 59-60, and revised this statement to be more accurate (lines 77-79).
>
> Lines 59-60:
>
> > "The light absorption of biomass burning aerosols are also largely dependent on the combustion conditions (Cheng et al., 2016a; Liu et al., 2014; Pokhrel et al., 2016; Saleh et al., 2014)."
>
> Lines 77-79:
>
> > "However, few studies have been conducted to gain a comprehensive understanding on the influence of combustion conditions on the chemical composition and light absorption of different BBOA fractions from agricultural residue burning."

*L110-111: Do authors mean that fuels were weighed before and after drying? If so please add this detail.*

> **Response**: Yes, the fuels were weighed before and after drying. We have added this detail and

revised this sentence as follows: "The moisture content was measured by weighing the fuels before and after drying the biofuels in the oven at 105℃ for 24 h." (lines 112-113)

*Table S1: There doesn't seem to be a consistent picture between MCE and the moisture content. For example, MCE ~0.77 was observed at all different moisture content values of the wheat. Please explain the reason for this variability. Because of this lack of obvious trend, I would not mention this in the conclusions either (L318-319)*

**Response**: We plotted the variations of MCE as a function of moisture contents (added in Figure S2 in the supporting information) and checked their correlation. The MCE generally decreased as moisture contents increased (Pearson correlation=0.73 at the 0.01 significance level). Burning conditions are not only influenced by biofuel moisture contents, but also biofuel structures (e.g., biomass sizes), combustion temperatures and ambient conditions, etc. (Chen and Bond, 2010; Lu et al., 2009; Sanchis et al., 2014) The variations in other factors could be the reasons for observing similar MCE values under different moisture contents. To exclude the influence of other factors, we thus conducted three parallel experiments under each condition (the same type of straw with the same level of moisture content, as listed in Table S1).

We added the explanation in lines 178-183, and revised the related sentence in the conclusion section (Line 330).

Lines 178-183:

"Similar MCE was also observed among wheat burning experiments with different levels of moisture contents (Table S1). This was because that MCE is not only influenced by biofuel moisture contents but also the variations of biofuel structures (e.g. size), burning temperatures or ambient conditions (Chen and Bond, 2010; Lu et al., 2009; Sanchis et al., 2014). We cannot completely exclude the differences of other factors between each parallel experiment, which was the reason for repeating each condition for three times in our experiment (Table S1)."

Line 342:

"The emission levels, compositions and light absorption of BBOA were influenced by the burning conditions."

Newly added Figure S2 in the supporting information:

[Figure]

Figure S2 Variations of MCE as a function of moisture contents. The Pearson correlation=0.73 at the 0.01 significance level.

References:

Chen, Y. and Bond, T. C.: Light absorption by organic carbon from wood combustion, *Atmos. Chem. Phys.*, 10, 1773-1787, 10.5194/acp-10-1773-2010, 2010.

Lu, H., Zhu, L., and Zhu, N.: Polycyclic aromatic hydrocarbon emission from straw burning and the influence of combustion parameters, *Atmos. Environ.*, 43, 978-983, 10.1016/j.atmosenv.2008.10.022, 2009.

Sanchis, E., Ferrer, M., Calvet, S., et al.: Gaseous and particulate emission profiles during controlled rice straw burning, *Atmos. Environ.*, 98, 25-31, 10.1016/j.atmosenv.2014.07.062, 2014.

*Eqn of Abs(l): why is absorbance at 700 nm subtracted from the absorbance at the wavelength of interest? Why should this be a relative absorbance? Shouldn't the absorbance at a specific wavelength be corrected for the background absorbance at the same wavelength while sampling only pure water?*

**Response**: Yes, the spectrum and absorption at a specific wavelength were determined and corrected relative to a reference cuvette which contained the same extraction solvent (water or methonal) during the measurement. $Abs_{700}$ (no absorption for BrC extracts) is subtracted from $Abs_{\lambda}$ to correct the systematic baseline drift of the of the instrument (Xie et al., 2017; Xie et al., 2019; Zhang et al., 2013; Zhu et al., 2018) (added in lines 139-140).

Lines 139-140:

"$A_{\lambda}$ is referenced to the $A_{700}$ to account for systematic baseline drift (Xie et al., 2019; Zhang et al., 2013)."

References:

Xie, M., Hays, M. D., and Holder, A. L.: Light-absorbing organic carbon from prescribed and laboratory biomass burning and gasoline vehicle emissions, *Scientific Reports*, 7, 7318, 10.1038/s41598-017-06981-8, 2017.

Xie, M., Chen, X., Hays, M. D., and Holder, A. L.: Composition and light absorption of N-containing aromatic compounds in organic aerosols from laboratory biomass burning, *Atmos Chem Phys*, 19, 2899-2915, 10.5194/acp-19-2899-2019, 2019.

Zhu, C. S., Cao, J. J., Huang, R. J., et al: Light absorption properties of brown carbon over the southeastern Tibetan Plateau, *Sci. Total Environ.*, 625, 246-251, 10.1016/j.scitotenv.2017.12.183, 2018.

*L178-180: The average EFs of corn are higher, but still considering the variabilities that were observed for both fuels, the difference isn't significant and beyond the observed variabilities.*

**Response**: Thanks for the reminding. We have removed this statement in the revised version.

*Fig. 3g: why not showing all the fits as in the other panels? Also, are the fits a double-sided*

*regression line, considering the uncertainties in the x and y values?*

**Response**: We have revised Figure 3g to show all the fits as in the other panels. Yes, all the fits are Pearson correlations considering the uncertainties in the x and y values. We have added the significant levels (2-tailed) in the revised figure.

Revised Figure 3:

[Figure]

Figure 3 Emission factors of $PM_{2.5}$, carbonaceous aerosols (OC, WSOC, $HULIS_C$ and EC) and EC/(OC+EC) ratios as a function of modified combustion efficiency (MCE). Corn and wheat burning emissions are denoted by red and blue colors, respectively. The r values in each panel are the Pearson correlations between emission factors and MCE for corn (blue), wheat (red) and the overall (black) burning experiments. The ** or * following the r value indicates the correlation is significant at the 0.01 level or 0.05 level (2-tailed).

*L233: what precluded the possibility of having burns with MCE>0.9 that's more representative of flaming conditions? I think some discussion should be provided.*

*Also it would be valuable to mention what the expected MCE in real world burns of agricultural residues are so readers get an idea of how applicable the results are and what values are most meaningful to be used in models.*

**Response**: We are afraid that the reviewer may misread our writing: "As the conducted experiments were mostly dominated by smoldering combustions (MCE=0.68-0.88) in this study, we CANNOT EXCLUDE the possibility that the EC emissions may be higher under flaming-dominated combustions (e.g. MCE>0.9)." (lines 252-253) Though we observed

flaming-dominated conditions during the initial period of low-moisture biomass burning experiment (as shown in Figure 1a), the whole combustion period was generally dominated by smoldering conditions based on the averaged MCE of 0.68-0.88 in this study.

Smoldering-dominated conditions, with expected MCE<0.9 or even lower, have been widely observed during the real world burns of agricultural residues in the agricultural area in China (Figure R1) and India (Figure R2). Thus, we believe our results, obtained under smoldering-dominated conditions (MCE=0.68-0.88), are applicable to the field or related model studies. Referring the observed or expected MCE or EC/OC ratios in specific study would help to select more suitable values in models. We have added the description in lines 170-172.

[Figure]

Figure R1 Intense straw burning in agriculture area (Anhui province, China) in China. (Wang, et al., 2017)

[Figure]

Figure R2 Post-harvest crop residue burning in northwest India. (IARI, 2012)

Lines 170-172:

"Smoldering-dominated conditions, with expected MCE<0.9 or even lower, have been widely observed during the combustion of agricultural residues in the field (IARI, 2012; Wang et al., 2017), thus the results in this study are applicable to the field or related model studies."

References:

Crop Residues Management with Conservation Agriculture: Potential, Constraints and Policy Needs, edited by: Institute, I. A. R., India, 2012.

Wang, Y., Hu, M., Lin, P., et al: Molecular characterization of nitrogen-containing organic compounds in humic-like substances emitted from straw residue burning, *Environ. Sci. Technol.*, 51, 5951-5961, 10.1021/acs.est.7b00248, 2017.

*Figure 4. There are some wheat burns for which the K+/OC and Cl-/OC ratios are highly variable; are all the burns from the same batch of fuel? Could this variability be explained by variable K and Cl content of the fuel itself?*

**Response**: Yes, all the wheat burns are from the same batch of biofuels. Thus, the differences in the K and Cl contents of biofuels among different experiments may be small.

Previous studies suggested that the contents of K and Cl released into smokes are related to elevated combustion temperatures during the biomass burning. The K and Cl begin to be released into the smokes when fire temperatures are higher than certain values (e.g. Temp.>600-700℃ for K, and Temp.> 200℃ for Cl), and the released proportion increase as the combustion temperatures further increased (Jensen et al., 2000; Knudsen et al., 2004). We checked the burning conditions of the two experiments with high $K^+$/OC and $Cl^-$/OC ratios. The moisture contents were 7% (the lowest moisture level in our experiments) and MCE of the two experiments were 0.77 and 0.79. Though the average MCE is not the highest, high fire temperatures were observed during the initial flaming combustion periods of the two low-moisture biomass burning experiments (added in Figure S4). The temperatures during these periods were much higher than the smoldering periods. We think that the higher ratios of released K and Cl were related to the elevated combustion temperatures during the initial flaming periods. We measured the fire temperatures using a sensor above the fires (as shown in Figure S1), the real combustion temperatures could be higher than the measured ones (e.g. higher than 600-700℃, which were suggested to be the K released temperatures during biomass burning). We have added the explanation in lines 266-268 and Figure S4.

Lines 273-275:

"The two wheat burning experiments (moisture content=7%) with higher $K^+$/OC and $Cl^-$/OC ratios (>0.5) than others were related to the higher combustion temperatures during the initial flaming periods of the burning experiments (Figure S4)."

Newly added Figure S4 in the supporting information:

[Figure]

Figure S4 Variations of measured fire temperatures during two wheat straw (moisture content=7%) burning experiments with MCE=0.77 and 0.79.

*L290-291: Since this paper has reported on MAE as well as EF of the different components of OC, it will be very valuable to combine the two results and present the EF of absorption to be able to more directly compare radiative impacts of WISOC, WSOC, and HULIS.*

**Response**: We estimated the radiation effects of WISOC, WSOC and HULIS relative to elemental carbon using a simplified model in the revised manuscript. Related descriptions have been added in section 2.3 (lines 149-155) and lines 313-314.

Lines 149-155 in section 2.3:

"The radiation effects of different BrC fractions (WSOC, HULIS and WISOC) relative to elemental carbon (EC, f) were estimated using a simplified model (Kirillova et al., 2014; Wu et al., 2020):

$$f = \frac{\int I_0(\lambda)\left\{1-e^{-(MAE_{BrC,365}\left(\frac{365}{\lambda}\right)^{AAE}\cdot C_{BrC}\cdot h_{ABL})}\right\}d\lambda}{\int I_0(\lambda)\left\{1-e^{-(MAE_{EC,870}(\frac{870}{\lambda})\cdot C_{EC}\cdot h_{ABL})}\right\}d\lambda} \tag{3}$$

where $MAE_{BrC,365}$ and $MAE_{EC,870}$ represent the MAE of different BrC fractions at 365 nm and MAE of EC at 870 nm. AAE is the AAE values of different BrC fractions obtained in this study, and the AAE of EC is set to 1. $C_{BrC}$ and $C_{EC}$ are the concentrations of BrC and EC, and $h_{ABL}$ is the height of atmospheric boundary layer (1000 m). $I_0(\lambda)$ represents the clear sky Air Mass 1 Global Horizontal solar irradiance (Levinson et al., 2010)."

Lines 313-314:

"The solar energy absorbed by biomass burning-emitted WISOC relative to EC (25%) among the wavelength range of 300-700 nm was higher than those of WSOC (10%) or HULIS (4%)."

L297-298, 338-339: I disagree; there are really two points that might be considered as outliers and without those, the MAE(365) vs MCE looks pretty flat. I suggest removing this statement.

**Response**: Thanks for the reminding. We carefully checked the correlations between $MAE_{365}$

and MCE again, and found their correlations were not significant at the 0.01 level (2-tailed) for either WSOC or HULIS. As suggested, we have removed this statement in the revised version.

**Suggested Edits:**

*L 33: remove observed in "..were also observed higher under. . ."*
    **Response**: Revised accordingly.

*L39 and 334: remove if in "if without considering the burning conditions. . ."*
    **Response**: Removed accordingly.

*L 61: Add ". . .was reported to be higher for more . . ."*
    **Response**: Added accordingly.

*L112: consider changing "weighted" to "weighed"*
    **Response**: Changed accordingly.

*L118: include the volumetric unit for both 10 and 5 units of water and methanol, respectively.*
    **Response**: Revised accordingly.

*L121: why did you use a smaller size filter for the WISOC fraction?*
    **Response**: We used a smaller size filter to extract the WISOC fraction for further analysis using HPLC-MS. It's just a test experiment this time due to the limited samples, and we plan to characterize the molecular compositions of water-insoluble BrC in our future studies.

*L129-L130: consider changing "minus. . ." to "difference between total OC and WSOC."*
    **Response**: Changed accordingly.

*Eqns. Consider adding equation numbers*
    **Response**: We have added the equation numbers in the revised manuscript.

*L174-175: I think I know what the authors try to say (in higher moisture fuel burns, some energy is first used to dry up the fuel and so the temperature is lower); however, as written the sentence is confusing. Consider rephrasing it.*
    **Response**: The sentence is now revised as: "In higher-moisture fuel burns, some energy released from the combustion is first used to dry up the higher moistures of the biofuels, thus the fire temperatures and burning efficiency were lower than those of the low-moisture biomass burning." (lines 195-197)

*L186 and 191: change negligible to "neglected"*
    **Response**: Revised accordingly.

*L200, 315: change "dominated" to "dominant"*
    **Response**: Changed accordingly. I think you may mean changing "negligible" to "neglected" in line 315 (line 339 in the revised version).

*L234-235: rephrase the beginning of the sentence; the structure is not correct*
    **Response**: The sentence is now revised as: "Though the EC emission factors did not show

obvious variation trends as a function of MCE, a positive correlation between EC/(OC+EC) ratios and combustion efficiency was observed (Figure 3g)." (lines 254-255)

*L237: data "are"...*

**Response**: Revised accordingly.

*L247: remove "that"*

**Response**: Removed as suggested.

*L272: consider changing "occupy" to "contribute to"*

**Response**: Changed accordingly.

*L271-273: the % contributions are for 300 nm and 400 nm, respectively? It's unclear when a range of 300-400nm is mentioned. Please clarify.*

**Response**: Thanks for pointing out the unclear statement. The % contributions here are for the HULIS$_C$ and high-polarity WSOC fractions, respectively. To be clear, the sentence is now revised as: "In the wavelength range of 300-400 nm, HULIS$_C$ and other high-polarity WSOC (WSOC-h=WSOC-HULIS$_C$) respectively contribute to 16%-28% and 1%-10% of the total BBOA absorption for corn burning, and 17%-29% and 12%-15% for wheat burning." (lines 294-296)

*L296, 338: change "as the decreasing of MCE" to "... as MCE decreases..."*

**Response**: Revised accordingly.